



# Evaluation of Changes in Some Physico-Chemical Properties of Bottled Water Exposed to Sunlight in Bauchi, Nigeria

Rose E. Daffi [1], Fwangmun B. Wamyil [2]

[1]Department of Civil Engineering, University of Jos, Jos, 930222, Nigeria
5 [2]Department of Civil Engineering, Kampala International University, Ishaka, Uganda

*Correspondence to*: Rose E. Daffi (rosedaffi@gmail.com)

**Abstract.** It is common for bottled water and other assorted drinks to be seen displayed outside stores and in the sun in most parts of Nigeria. The country is mostly hot year-round and over the course of the year temperatures can rise to as high as 40ºC around March-April in the study area. The leaching effect of chemicals from polyethylene terephthalate (PET) bottled water 10 was investigated for five (5) commercially available bottled water brands. Temperature, pH, Antimony, Bisphenol A and Nitrate levels were measured on day zero, 14 and 28 for control samples and samples exposed to direct sunlight, using destructive sampling technique. The study found that pH for brands B and D were lower than the Nigerian Standard for Drinking Water Quality (NSDWQ) and United States Environmental Protection Agency (US EPA) regulations at day zero. The control sample for Brand A maintained pH within the guideline values for 0-28days while all Brands exposed to sunlight 15 for 14 and 28days had lower values than specified. Antimony was not detected in brands A, B and E in the baseline measurement at day zero while brands C and D had low values; after 28days all the control samples still had Antimony levels within the US EPA standard. Meanwhile all the samples exposed to sunlight exceeded US EPA standard levels at 14 and 28days except brand A which was within limit at 14days with value of 4.59µg/L. All control and exposed samples were below the European Union Drinking Water Directive (EU DWD) total daily intake (TDI) of Bisphenol A (0.05mg/kg body 20 weight/day). Values obtained for Nitrate showed that all control samples did not exceed the US EPA guideline level for Nitrates in drinking water for 0, 14 ad 28days while three (3) of the samples, Brands C, D and E, exceeded the guideline level at 28days. Expose of bottled water to sunlight was seen to impair the quality of the water for consumption. It is recommended that regulators and practitioners drive implementation of proper storage/retailing of bottled water products and improve legislation on manufacture of plastics for food contact products.

25 **Keywords:** Antimony (Sb), Bisphenol A (BPA), Nitrates ($NO_3$), Polyethylene terephthalate (PET), pH, temperature, sunlight exposure





# 1 Introduction

Water is an important resource to the wellbeing and development of plant and animal life. For humans, water constitutes over 50% of the body weight (Salvato et al., 2004; Weiner and Matthews, 2003). Water is a medium for transporting nutrients and removing wastes generated from cells that aid the body in functioning optimally; it is recommended that an adult takes between 2-3litres (sedentary) or 4.5litres (if engaged in physical activity) to support hydration(WHO, 2005). Various sources of water that can be used (potable water) for carrying out human activities abound in the environment. They include rainwater, surface water and groundwater (Mays, 2011). Bottled water is classified a consumer food product and consist of spring, purified, mineral, sparkling, artesian and well water which are to be carefully processed to meet regulatory standards (IBWA, 2020). Bottled water is a good option compared to other beverages especially those that contain high sugar content.

Historical records show the introduction of for sale bottled water to the 17th century in the United Kingdom (Holy Well Bottling Plant) when water from mineral Springs were thought to possess therapeutic and healing properties and sold as remedies for ailments (Malvern Beacon, n.d.; Mitte Team, 2019). When Johann Jacob Schweppe (1783) discovered how to carbonate water, the 'fizzy' quality of plain water gave a considerable competition to mineral water (Mitte Team, 2019; Schweppes-Heritage, n.d.). Lowering cost of glass and techniques of bottling increased the popularity of bottled water and carbonated drinks in the United States in the 19th century. Sales were common up till the early part of the 20th century when Chlorination was discovered to make tap water safer, causing a decline in the choice of bottled water (Hurly, 2016; National Center for Environmental Health et al., 1999).

The introduction of polyethylene terephthalate (PET) as a replacement for glass in the 1970-80s caused a reduction in cost of manufacturing bottled drinks (including water) (Leigh, 2011; Mitte Team, 2019; Parker, 2019). PET is a clear, strong and light weight plastic (thermoplastic polyester resin) used in producing packaging materials for food, beverages, cosmetics, water, photographic films; and in making apparels (International Life Sciences Institute, 2000; PETRA, 2015; US EPA, 1995). PET was first synthesized in the 1940s by DuPont chemists (PETRA, 2015). The processing of PET involves mixtures of ethylene glycol (EG) and terephthalic acid (TPA) or dimethyl terephthalate (DMT) to form chain of polymers; the output is extruded, cooled and cut into small pellets; the pellets are moulded into any shape by heating them into a molten liquid (PETRA, 2015; US EPA, 1995). PET can be recycled by breaking it down into its constituents and using same to make new PET materials, unfortunately large amounts of this product still find its way to landfills, open dumps and improperly disposed waste where it breaks down to micro-plastics and nano-plastics ultimately finding its way to the marine ecosystem with deleterious environmental effects (PETRA, 2015; US EPA, 1995).

Drinking water quality is a major risk factor for diseases and established standards set to ascertain the suitability of water for intended use. These standards can be broadly categorized as physical, chemical, biological and radiological (Davis, 2010; Gaur, 2008; Spellman, 2003).





The physical properties of water are colour, turbidity, temperature, taste and odour. Temperature is the amount of heat contained in water and it determines how palatable the taste, colour and corrosion of water; warmer water will absorb inorganic

and other contaminants making it less palatable than cooler water. 16ºC is generally acceptable in the United States (Gaur, 2008; Salvato et al., 2004; Spellman, 2003; WHO, 2017).

The chemical properties of water are pH or the presence of chemicals like arsenic, Iron, Lead, Sodium, Zinc or other toxic organic or inorganic substances. pH is the acidity or alkalinity of water determined by measure of Hydrogen (H+) ion concentration, with "acidic" having high Hydrogen (H+) ion concentration and "alkaline" having high concentration of

Hydroxyl (OH-) ions. The pH scale ranges from 0-14, with 7 being neutral at 0ºC (Gaur, 2008; Spellman, 2003). Some chemicals are essential to human and animals in trace amounts, but prolonged exposure in higher amounts can be dangerous to human health. Chemicals can occur naturally from water source; whereas others are as a result of human activities (industrial-mining and human dwellings); agricultural activities (fertilizer and pesticide application); water treatment (supply lines, coagulants); pesticides (public health use); or containing vessels where water is stored (plastic bottles) (WHO, 2017).

Regulations specifically aim at ensuring the deleterious effects of the chemicals are avoided. Some chemicals called disinfection products (DBP) get in water as a result of disinfection (chlorination) in water treatment process. They include the trihalomethanes (THMs) and haloacetic acids (HAAs) which are the main DBPs worth noting (WHO, 2017). Other chemicals like Antimony, Bisphenol A and Nitrates can be present in drinking water as a result of plastic bottle feeding of infants and other plastics including bottled drinking water (US EPA, 2010; WHO, 2017). For example, Antimony found in Antimony

trioxide is used as a flame retardant or in polycondensation when PET is produced (Bach et al., 2014; WHO, 2003a).

The biological characteristics of water includes presence of pathogenic organisms-viruses, protozoa, helminths, bacteria which can cause illnesses such as typhoid, diarrhoea, tape worms, round worms. The presence of *Escherichia coli* (*E. coli*), *Enterobacter clocae, Citrobacter freundii,* common in stool and sputum of warm-blooded animals including human proves the contamination of water by stool (Salvato et al., 2004; SON, 2007; Weiner and Matthews, 2003; WHO, 2017). Some of the

organism however can grow in water distribution systems, reproduce as a result of warm temperatures and be inhaled as aerosols (amoebae *Naegleria fowleri* and *Acanthamoeba* spp.) (WHO, 2017). The target for most regulations is to ensure that no pathogenic organism is present in water (FDA, 2016; WHO, 2017).

Radiological standards deal with water in contact with radioactivity. Parameters like alpha, beta particles, Radium-226, 228 and Uranium have standards set for drinking water to ensure no untoward health effects to human wellbeing when consumed

(Davis, 2010). Radiation occurs naturally (accounting for almost 80% of dose from all radiation sources), from medical diagnosis (accounts for 19.6%) and from man-made sources (accounts for 0.4%) (WHO, 2017).

A summary of the drinking water (and bottled water) quality standards available as issued by major institutions for physical, chemical, biological and radiological characteristics; and suspected health effects are presented in Table 1.

There is increasing concern for the leaching of chemicals from containers (including PET) as a result of increasing ambient temperatures (University of Florida Institute of Food and Agricultural Sciences, 2014; Zanolli, 2019). In most parts of Nigeria, including the study area, bottled water and other assorted drinks for sale are displayed outside stores and in the sun for days and weeks or hawked in traffic by the roadside until they are sold. The country is mostly hot year-round. This study seeks to investigate the changes in levels of specific contaminants (Antimony, Bisphenol A, Nitrates, pH) in five (5) commercially available market brands of bottled water continually exposed to sunlight for $0 - 28$ days in Bauchi, Bauchi State and the effects on the quality of the water. The study area is a metropolitan city and capital of Bauchi State in Nigeria. Over the course of the year temperatures can rise to as high as 40ºC around March-April in the study area. The study was carried out between December, 2019 to January, 2020 when average maximum temperature was 31ºC. (World Weather Online, 2020)

## 2 Materials and Methods

### 2.1 Reagents

Reagent grade analytical chemicals and deionized water were used for preparing all solutions.

### 2.2 Sampling of The Bottled Water

Chemical analysis of the water contained in the plastic bottles were carried out in order to investigate whether the bottles leach chemical(s) to the bottled water under heat (source: sun) and the level of the chemical concentration over time (28days) in the bottled water. Five (5) commercially available brands of bottled water (obtained directly from the suppliers) were taken as samples and tested (for pH, Antimony, Bisphenol A, Nitrates and Temperature) immediately after wards to establish the baseline parameters. Samples (labelled A to E) were exposed under direct sunlight and control samples were kept in storage to replicate non-exposure to direct sunlight. Sample testing was carried out at 14 and 28 days between December 2019 (Ambient: Max temp = 31ºC, Avg. temp = 16ºC and Min Temp = 16ºC) and January 2020 (Ambient: Max temp = 31ºC, Avg. temp = 24ºC and Min temp = 14ºC) (World Weather Online, 2020) using a destructive sampling technique.

### 2.3 Procedures Used in Measuring Parameters

pH and Temperature were evaluated using standard methods according to (APHA, 2018c) and (APHA, 2018a) respectively. Antimony was measured according to 3110 Metals by Atomic Absorption Spectrometry (APHA, 2018b). The presence of Nitrate was determined using the colorimeter (sulphanilamide method) according to 4500-NO3− Nitrogen (Nitrate) (APHA, 2018d). Bisphenol A was measured using High-performance liquid chromatography (HPLC) according to 6810 Pharmaceuticals and Personal Care Products (APHA, 2018e).



## 2.4 Method of Data Analysis

The data in this study was statistically analysed using R (3.6.3) (R Core Team, 2020) programming (in RStudio Version 1.2.5033). The method of data presentation of tables and charts were done in comparison to the water quality standards set out by WHO (2006) guidelines for drinking water quality, US EPA (2009) National Primary Drinking Water Regulations and the Nigerian Standard for Drinking Water Quality (NSDWQ).

## 3 Results and Discussion

### 3.1 pH

The pH values obtained from the tests carried out for the five brands of bottled water are as given in Fig 1. The baseline (day zero) pH values for all brands ranged between 5.25 and 7.4. For the control sample at 14 day and 28 day, there was a general decrease of pH implying increase in acidity in the 5 brands. Brand A showed approximately 3.8% and 5.8% reduction in pH in day 14 and 28; Brand B-3.6% and 7.3%; Brand C-5.4% and 9.2%; Brand D-13% and 20% (highest reduction); and Brand E- 2.8% and 7% respectively. The final pH values at the 28 day for Brands A, B, C, D and E were 6.97, 5.93, 6.45, 4.17 and 6.07 respectively (Fig. 1**Error! Reference source not found.**). Samples exposed to direct sunlight had temperatures of 47°C and 56°C on day 14 and day 28 respectively with Brand A showing 15% and 20% reduction in pH at day 14 and day 28; Brand B-23.4% and 46.7%; Brand C-16% and 39%; Brand D-19.8% and 49%; and Brand E- 27% and 54.8% respectively. The final pH values at the 28 day for Brands A, B, C, D and E were 5.92, 3.41, 4.32, 2.67 and 2.95 respectively.

The baseline pH values all fall within range of standard (NSDWQ and US-EPA: 6.5-8.5) except for Brands B and D. For the control samples at the 28 day only Brand A had pH at regulatory level (pH of 6.97). The sample exposed to sunlight had a lower pH (higher acidity) for each Brand and also as the days of exposure to sunlight increased. This is similar to research by (Muhamad et al., 2011), where samples were exposed to sunlight for 5 days (there was a decrease in pH with mean max temperature between 41°C to 47°C). However, the result from this study varies from investigation by (Akharame et al., 2018), where the pH values after the 28 day were all within the WHO and NSDWQ regulation levels. WHO has highlighted health effects of pH<4 to include eye redness and irritation and for pH<2.5 damage to epithelium (WHO, 2003b). The result suggests that sunlight exposure (temperature) affects pH of bottled water.

### 3.2 Antimony

The values for antimony obtained from the tests carried out for the 5 Brands of bottled water are as given in Fig 2. In the baseline measurement (day zero), Antimony was not detected in Brands A, B and E while Brands C and D had values of 0.095 µg/L and 0.35 µg/L respectively. At day 14, the control samples had a range of Antimony values between 1.53µg/L and 4.35µg/L respectively (Fig. 2). On day 28, Antimony levels for the control samples had a range of 2.87µg/L and 5.41µg/L.



This corresponded to an 88%, 45%, 21%, 24% and 37% increase from day 14 to day 28 for the control samples. The samples exposed to sunlight had a range of 4.59µg/L and 11.41 µg/L on day 14; and 8.17 µg/L and 14.49 µg/L on day 28. This corresponded to about 78%, 58%, 34%, 27% and 8% increase from day 14 to day 28 of being exposed to sunlight.

The increase in level of Antimony in samples exposed to sunlight after the 28 day is higher than in the experiments carried out by (Bach et al., 2014). In that research, water samples had increase in Antimony levels of about 140% (1µg/L) after the 10th
day of exposing mineral water (non-carbonated) while for ultra-pure the increase after the 10th day was 50%. The increase in levels of Antimony could be attributed to effects of pH, temperature and UV radiation (Fig. 2) from the sun, supported by experiments from (Al-Khatim et al., 2019; Chapa-Martínez et al., 2016; Dogan and Cebi, 2019; Zmit and Belhaneche-Bensemra, 2019) which identified the significance of pH, temperature and days of exposure in the amount of Antimony leached into bottled water. The high level of the Antimony values can also be attributed to the type of plastics used by the different
brands as identified by (Koyuncu and Alwazeer, 2019) who suspected reason for high Antimony levels in milk contained in PET bottles results from type and production processes employed in making containers. Qiao et al. (2018) also recorded higher values of Antimony leached in drinking water bottles exposed to high temperatures. Notably, all control samples did not exceed the US-EPA level of 6µg/L (Fig. 2), while in the samples exposed to sunlight all exceeded the US-EPA values after the 14th day though they did not exceed the WHO regulation of 20µg/L.

**3.3 Bisphenol A (BPA)**

The values of Bisphenol A obtained from the tests carried out for the 5 brands of bottled water are as given in Fig 3. The baseline results for both control and sunlight exposed samples were below limit of detection as seen in Fig. 3. For the control sample on day 14 there was increase in level of BPA to 3.61µg/L, 3.92 µg/L, 4.1 µg/L, 5.4 µg/L and 3.46 µg/L for Brands A, B, C, D and E respectively. At the 28th day, there were 14.7%, 14.5%, 20.7%, 25.7% and 24.6% increase respectively from
day 14. The samples exposed to sunlight showed BPA levels of 8.64 µg/L, 9.34 µg/L, 10.1 µg/L, 13.42 µg/L and 8.59 µg/L for Brands A, B, C, D and E respectively at day 14. There were also corresponding increases on day 28 to 10.34 µg/L, 11.95 µg/L, 13.59 µg/L, 15.48 µg/L and 9.72 µg/L showing 19.7%, 28%, 34.6%, 15.4% and 13.2% increase respectively.

BPA has come under debate about its effect on health, especially for infants and children leading to the restriction in items directly used by them in the US (US EPA, 2010), though observations (and petitions) leading to further research has shown
that there is conflicting data correlation to its (BPA) negative effect on reproductive health (hormonal disruptor) leading to no definite legislation/regulation by either WHO or US agencies (WHO and FAO, 2010). Restriction on intake per body weight exists such as tolerable daily intake (TDI) of 0.05mg/kg body weight (bw)/day by European Food Safety Authority (EFSA, 2008). In this study, the highest increase in BPA level for control and sunlight exposed sample was 6.97µg/L and 15.48 µg/L respectively for Brand D. Using an average (crude) weight of men and women in the US for 2015/2016 as 89.7kg and 77.3kg
(Fryar et al., 2018) and an average daily intake of 4.5litres for a physically active person (WHO, 2005), the daily intake for



this study is calculated as 0.772µg/kg (bw)/day and 0.901µg/kg (bw)/day for men and women respectively using Eq. (1). For a child aged 3-6months assuming the same amount of BPA is transferred through water intake of 1.171L/day (95th Percentile) (US EPA, 2011), the child will have a BPA level of 18.112µg/day (without dividing by average weight) or 3.012µg/kg (bw)/day for a child weighing 6kg Eq. (1).

$$\text{Daily intake} = A * \frac{B}{C} \tag{1}$$

[Where: Daily intake $of\ Bisphenol\ A\ for\ this\ study$ $(µg/kg(bw)/day)$, A = Max Bisphenol A level in this study $(µg/L)$, B = Water standard daily intake $(L/day)$, C = Average body weight $(kg\ (bw))$ ]

The TDI stipulated by EFSA (2008) is more than those calculated for men, women and children (0.772µg/kg (bw)/day, 0.901µg/kg (bw)/day and 3.012µg/kg (bw)/day respectively), implying that there is no significant risk to a person based on
crude average weight (male, female, children) and recommended average daily water intake (for physical activity).

### 3.4 Nitrate

The values for Nitrate obtained from the tests carried out for the 5 brands of bottled water are as given in Fig 4. The baseline (day zero) Nitrate levels were 0.22mg/L, 1mg/L and 0.05mg/L for Brands B, C and D respectively; while Brands A and E were not detected as seen in Fig. 4. The control samples on day 14 had Nitrate level of 0.54mg/L, 1.21mg/L, 2.41mg/L
0.93mg/L and 0.62mg/L for Brands A, B, C, D and E respectively. After the 28th day, the Nitrate level increased by 142.5%, 100.8%, 60.2%, 101.1% and 151.6% for Brands A, B, C, D and E respectively corresponding to 1.31mg/L, 2.43mg/L, 3.86mg/L, 1.87mg/L and 1.56mg/L. The samples exposed to sunlight showed levels of Nitrate after the 14th day as 4.54mg/L, 6.12mg/L, 7.71mg/L, 9.38mg/L and 6.79mg/L for Brands A, B, C, D and E respectively. An increase was noticed after the 28th day of 66.1%, 56.2%, 51%, 63.2% and 56.6% corresponding to 7.54mg/L, 9.56mg/L, 11.64mg/L, 15.31mg/L and
10.63mg/L for Brands A, B, C, D and E respectively.

The results showed that all levels of Nitrate in the control did not exceed the levels for US EPA (10mg/L) or that of the NSDWQ, WHO or EU (50mg/L). For samples exposed to sunlight, significant increase was noticed in the level of Nitrate as the study progressed leading (after the 28th day) to Brands C, D and E exceeding the US EPA (2009) regulation for Nitrates in drinking water. Another study (Muhamad et al., 2011), also recorded increase in Nitrate as temperature increased, though, the
Nitrate levels were not as high as in this study. Lastly, all Nitrate values (100%) in this study still fell below 50mg/L standards of WHO (2006) and NSDWQ (2007).





## 4 Conclusion

Leaching of chemicals from PET bottles have been investigated by (Abboudi et al., 2018; Al-Khatim et al., 2019; Zmit and Belhaneche-Bensemra, 2019). This study found that for each of the parameters the following conclusions apply:

1. The pH for commercially available brands of bottled water is affected by sunlight exposure and based on the conditions of storage and sale will be a potential source of ill health when consumed.

2. Sunlight exposure accelerates the leaching of Antimony and Nitrates overtime as reported by similar studies. This is in addition to the characteristic of plastic bottled containers used for storing bottle water.

3. Though sunlight exposure increases the leaching of Bisphenol A, it does not translate to any significant threat to health when consumed in line with current legislation by European Food and Safety Agency (EFSA).

4. The level of Nitrate leached as a result of sunlight exposure is within the limit set by NSDWQ and WHO.

We believe that to mitigate the potential negative impact of sunlight exposure on bottled water quality, there is a need for practitioners and regulators in Nigeria-NAFDAC (National Agency for Food and Drug Administration and Control) and FCCPC (Federal Competition and Consumer Protection Commission) to advocate and enforce the importance of proper storage of bottled water products by retailers and sellers as per the guideline proffered by NAFDAC -"products (water) should be stored in a cool and dry place away from sunlight" as reflected in (NAFDAC, 2018). The standards for the manufacturing of PET plastics for used as bottled water containers will need to be reviewed by the SON (Standards Organization of Nigeria) possibly in a similar manner as employed in the European Union (European Commission, 2011).

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

**Table 1: Regulated Standards of Selected Physical, Chemical, Biological and Radiological Parameters**

| Parameter | NSDWQ[a] | WHO[b] | US EPA[c] | EU-DWD[d] | Suspected Health Effects[a,b,c,d,e] |
|---|---|---|---|---|---|
| Turbidity | 5NTU | | 5NTU | 4NTU | Indicates inefficient water treatment process/distribution system. |
| Temperature | Ambient | - | - | | High temperature enhances microbial growth and problems associated with taste, odor, color and corrosion[a] |
| Color | 15TCU | - | 15 color units | 20mg/L Pt/Co | Indicates unsafe water/presence of chemical or biological impurities. |
| pH | 6.5 – 8.5 | - | 6-8.5 | | Acidic pH corrodes conduits & leach chemicals into water, irritate eyes. |
| Antimony (Sb) | - | 20µg/L | 6µg/L | 5.0µg/L | Affects blood (increase in cholesterol, decrease in sugar) |
| Arsenic (As) | 0.2mg/L | 10µg/L | 10µg/L | 10µg/L | Affects Skin & nervous system, increases cancer risk |
| Lead (Pb) | 0.01mg/L | 10µg/L | 5µg/L | 25µg/L | Affects central and peripheral nervous system, cancer & kidney damage |
| *E.coli* | 0cfu/100mL | | 0/100mL | 0/100mL | Gastroenteric disease |
| Total coliforms | 10cfu/100mL | | 0/100mL or <5% incidence/month | 0/100mL | Gastroenteric disease |
| Alpha particles (α)(activity) | 0.1Bq/L | 0.5Bq/L | 15pCi/L | | Increases Cancer risks |
| Beta particles (β) (activity) | 0.1Bq/L | 1Bq/L | 4 millirems per year | | Increases Cancer risks |
| Radium (Ra)-226 + Radium (Ra)-228 | - | 1Bq/L | 5pCi/L | | Increases Cancer risks |
| Uranium | - | 30µg/L | 30µg/L | | Increases Cancer risks |

[a](SON, 2007), [b](WHO, 2017), [c](US EPA, 2001, 2009), [d](Drinking Water Inspectorate, 2017), [e](Government of Canada, 2020)

[NSDWQ = Nigerian Standard for Drinking Water Quality, WHO = World Health Organization, US EPA = United States Environmental Protection Agency, EU-DWD = European Union Drinking Water Directive, TCU = True Color Units, NTU = Nephelometric Turbidity Units, Bq/L = Becquerel/Liter, pCi/L = picocuries per Liter, µg/L=micrograms/Liter, mg/L=milligrams/Liter, mL=milliliters, Pt/Co = Platinum-Cobalt scale]

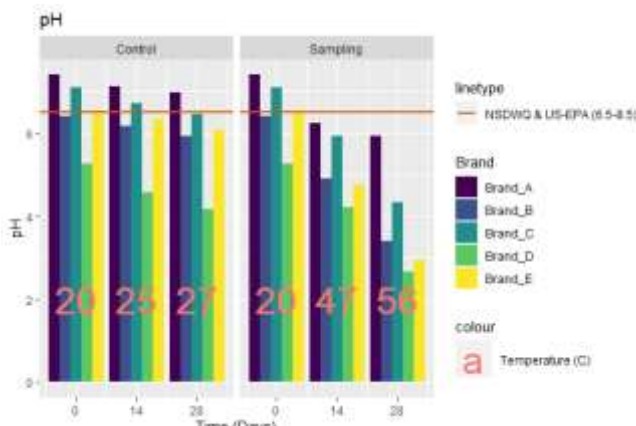

**Figure 1: pH Values for Control Group and Sunlight Exposure Group.**

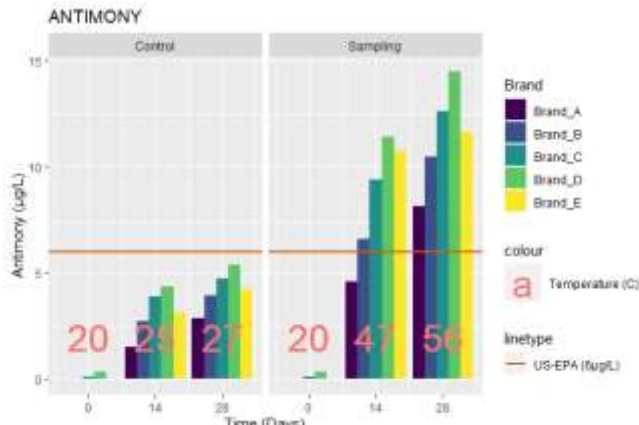

**Figure 2: Antimony Values for Control Group and Sunlight Exposure Group.**

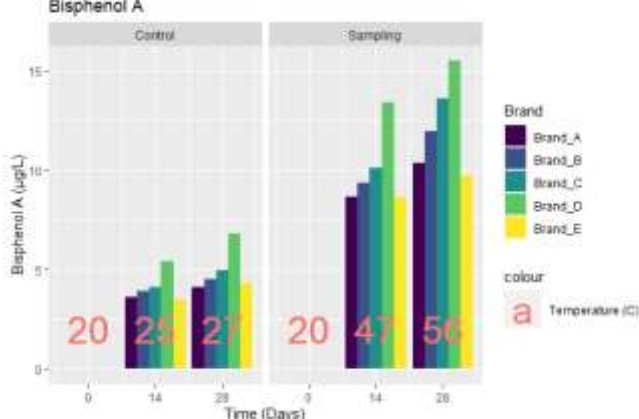

**Figure 3: Bisphenol A Values for Control Group and Sunlight Exposure Group.**



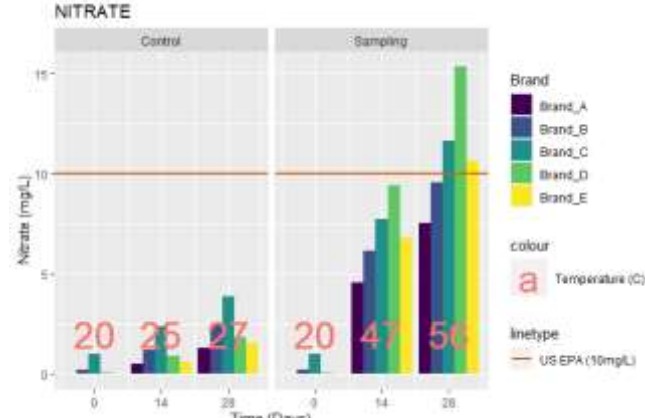

**Figure 4: Nitrate Values for Control Group and Sunlight Exposure Group.**