# Peer review of "Evaluation of Changes in Some Physico-Chemical Properties of Bottled Water Exposed to Sunlight in Bauchi, Nigeria"

_Drinking Water Engineering and Science, 2020_

## Referee Comment (RC1) · Anonymous Referee #1 · 29 Sep 2020

I believe that the present paper contains interesting results. Therefore, I think the manuscript could be accepted for publication after the following minor issues are addressed. Concrete comments are shown as follows. a) Abstract Line 22-24 – I would suggest to leave out the recommendation since more information is needed (especially regarding characterization of the bottles, environmental conditions such as pressure, humidity. . .). b) Introduction 1) General observation – Introduction is too long with irrelevant information in this case such as line 70-72 "Some chemicals called disinfection products (DBP) get in water as a result of disinfection (chlorination) in water treatment process. They include the trihalomethanes (THMs) and haloacetic acids (HAAs) which are the main DBPs worth noting (WHO, 2017)." 2) Sentences should be shorter and

to the point. For example: "PET can be recycled by breaking it down into its constituents and using same to make new PET materials, unfortunately large amounts of this product still find its way to landfills, open dumps and improperly disposed waste where it breaks down to micro-plastics and nano-plastics ultimately finding its way to the marine ecosystem with deleterious environmental effects (PETRA, 2015; US EPA, 1995)." 3) Remove line 35 "Bottled water is a good option compared to other beverages especially those that contain high sugar content." 4) What is the relevance of this paragraph on the topic: "The biological characteristics of water includes presence of pathogenic organisms-viruses, protozoa, helminths, bacteria which can cause illnesses such as typhoid, diarrhoea, tape worms, round worms. The presence of Escherichia coli (E. coli), Enterobacter clocae, Citrobacter freundii, common in stool and sputum of warm-blooded animals including human proves the contamination of water by stool (Salvato et al., 2004; SON, 2007; Weiner and Matthews, 2003; WHO, 2017). Some of the organism however can grow in water distribution systems, reproduce as a result of warm temperatures and be inhaled as aerosols (amoebae Naegleria fowleri and Acanthamoeba spp.) (WHO, 2017). The target for most regulations is to ensure that no pathogenic organism is present in water (FDA, 2016; WHO, 2017)." 5) In the introduction it is mentioned: "Over the course of the year temperatures can rise to as high as 40oC around March-April in the study area. The study was carried out between December, 2019 to January, 2020 when average maximum temperature was 31oC. (World Weather Online, 2020)"

However in the 2.2 paragraph it is mentioned: "December 2019 (Ambient: Max temp = 31oC, Avg. temp = 16oC and Min Temp = 16oC) and January 2020 (Ambient: Max temp = 31oC, Avg. temp = 24oC and Min temp = 14oC) (World Weather Online, 2020) using a destructive sampling technique." Could you explain the difference in average temperatures? Was the temperature continuously monitored during 28 days? How could you explain these data being representative for temperatures of 40C while having experiments done during average of 16 and 24C? c) Results and discussion 6) Different bottles either control samples or samples exposed to sunlight are showing changes to

pH, antimony, BPA and nitrate concentrations. However the explanation and link is missing to why is there a different decrease in for example pH. What is the structure of the bottles itself and what is the water matrix. What is effecting one sample to have pH decrease of 3.6 % while other 20%.

---

## Author Comment (AC1) · 2 Oct 2020

Dear Referee, thank you for considering our paper publishable with minor corrections. Your comments on some aspects of the paper are well noted. A revised version will include edits on your areas of observation. Please find as follows our responses to your comments: a) Abstract Line 22-24 – I would suggest to leave out the recommendation since more information is needed (especially regarding characterization of the bottles, environmental conditions such as pressure, humidity: : :). Author Response: The recommendation has been removed from the Abstract. b) Introduction 1) General observation – Introduction is too long with irrelevant information in this case such

as line 70-72 "Some chemicals called disinfection products (DBP) get in water as a result of disinfection (chlorination) in water treatment process. They include the trihalomethanes (THMs) and haloacetic acids (HAAs) which are the main DBPs worth noting (WHO, 2017)." Author's Response: The intention of including the paragraph was to inform the readers about the different aspects of water treatment and possible effects of disinfectants on chemical composition of the water. However, the section has been removed and the introduction reviewed. 2) Sentences should be shorter and to the point. For example: "PET can be recycled by breaking it down into its constituents and using same to make new PET materials, unfortunately large amounts of this product still find its way to landfills, open dumps and improperly disposed waste where it breaks down to micro-plastics and nano-plastics ultimately finding its way to the marine ecosystem with deleterious environmental effects (PETRA, 2015; US EPA, 1995)." Author's Response: Sentences have been modified. 3) Remove line 35 "Bottled water is a good option compared to other beverages especially those that contain high sugar content." Author's Response: Sentence has been removed. 4) What is the relevance of this paragraph on the topic: "The biological characteristics of water includes presence of pathogenic organisms-viruses, protozoa, helminths, bacteria which can cause illnesses such as typhoid, diarrhoea, tape worms, round worms. The presence of Escherichia coli (E. coli), Enterobacter clocae, Citrobacter freundii, common in stool and sputum of warm-blooded animals including human proves the contamination of water by stool (Salvato et al., 2004; SON, 2007; Weiner and Matthews, 2003; WHO, 2017). Some of the organism however can grow in water distribution systems, reproduce as a result of warm temperatures and be inhaled as aerosols (amoebae Naegleria fowleri and Acanthamoeba spp.) (WHO, 2017). The target for most regulations is to ensure that no pathogenic organism is present in water (FDA, 2016; WHO, 2017).") Author's Response: Though the paper focuses on physico-chemical changes in water and no biological test was carried out directly in the study, we believe that the paragraph will inform the readers about the biological aspects of drinking water quality worthy of note. But the paragraph has been removed for lack of relevance to the topic. 5) In the intro-

duction it is mentioned: "Over the course of the year temperatures can rise to as high as 40oC around March-April in the study area. The study was carried out between December, 2019 to January, 2020 when average maximum temperature was 31oC. (World Weather Online, 2020)". However, in the 2.2 paragraph it is mentioned: "December 2019 (Ambient: Max temp = 31oC, Avg. temp = 16oC and Min Temp = 16oC) and January 2020 (Ambient: Max temp = 31oC, Avg. temp = 24oC and Min temp = 14oC) (World Weather Online, 2020) using a destructive sampling technique." Could you explain the difference in average temperatures? Was the temperature continuously monitored during 28 days? How could you explain these data being representative for temperatures of 40C while having experiments done during average of 16 and 24C? Author's Response: There was error in typing in the average temperature for December 2020 and it has been corrected accordingly. Secondly, for the water samples the temperature was not continuously measured for 28days, it was measured at the point of sample analysis. Though the study is not representative of temperatures of 40oC, but we believe it is still relevant in understanding the effect of exposing bottled water products to direct sunlight, which is all through the year. c) Results and discussion 6) Different bottles either control samples or samples exposed to sunlight are showing changes to pH, antimony, BPA and nitrate concentrations. However, the explanation and link are missing to why is there a different decrease in for example pH. What is the structure of the bottles itself and what is the water matrix? What is affecting one sample to have pH decrease of 3.6 % while other 20%. Author's Response: There was no study carried out on the structure of the bottles or water matrix to understand and discuss their effect on the pH levels. However, information we have show that for Sample A raw water is sourced from a spring while the other samples (B-E) are sourced from deep boreholes. Depending on the source, dissolution of minerals (for example limestone which tend to increase pH by dissolving carbonates, bicarbonate and hydroxide compounds; or dissolved carbon dioxide which will decrease the pH over time. The process of water treatment and the minerals added can also account for the change of pH in control samples. The discussion of pH results has been modified in the paper.

Thank you.

Interactive
comment

---

## Referee Comment (RC2) · Anonymous Referee #2 · 26 Nov 2020

It is an interesting small study on the deterioration of the bottled water in Nigeria. The paper is well written. General comments: - Now much emphasis is given to the compliance with EPA and WHO regulations. More important is the effect of PET bottles and sunlight on the quality. - The abstract is too long with too specific details on compliance with regulations - Introduction has too many generalities, be more specific - Not clear why pH and nitrate are chosen as indicators for PET leaching. Consider at least removing nitrate as indicator. - Results frequently lacks critical discussion on mechanisms supported by literature. - Avoid recommendations in the paper. Specific comments - Line 12-15: delete sentence - Line 22-24: delete sentence - Line 28-33: delete sentence - Line 35: delete sentence - Line 58-69: delete sentences - Line 70-72:

[Figure]

delete sentences - Line 76-86: delete sentences - Line 88: Table 1 is not a representative summary of drinking water standards - Line 101: The = the - Line 105: after wards = afterwards - Line 121-139: this lacks a critical discussion. Why pH changes over time and why is it of importance? Drinking water standards are not relevant in bottled water. . . - Line 148: is = was - Line 149: by (Bach et al., 2014) = by Bach et al. (2014) - Line 149: not clear. Increase in study of Bach was lower but 140% (which is higher) - Line 168-171: delete sentences - Line 176: is = was - Line 186-201: this lacks a critical discussion. Why nitrate changes over time? Consider deleting - Line 205-207: adjust depending on changes in the document - Line 211: adjust depending on changes in the document - Line 212-218: delete sentences
* * *

---

## Author Comment (AC2) · 21 Dec 2020

Dear Referee, our appreciation for your meticulous observations which are duly noted and will be included in the revised version of the paper. Kindly find our responses to your comments:

[a.] General comments: - Not much emphasis is given to the compliance with EPA and WHO regulations. More important is the effect of PET bottles and sunlight on the quality.

Response: comments are kindly noted.

[b.] The abstract is too long with too specific details on compliance with regulations

Response: Abstract has been updated as suggested in addition to RC1 comments.

[c.] Introduction has too many generalities, be more specific.

Response: Introduction has been updated in addition to RC1 comments.

[d.] Not clear why pH and nitrate are chosen as indicators for PET leaching. Consider at least removing nitrate as indicator.

Response: Though pH and Nitrate are not direct indicators for PET leaching, we decided to include them as they are parameters that were indicated on the products and we wanted to study/ensure that they were within acceptable range when exposed to sunlight. We have updated the sentences line in the paper (lines 92-95):

"This study seeks to investigate the changes in levels of specific parameters (Antimony, Bisphenol A, Nitrates, pH) in five (5) commercially available market brands of bottled water continually exposed to sunlight for 0 – 28days in Bauchi, Bauchi State and the effects on the quality of the water."

[e.] Results frequently lacks critical discussion on mechanisms supported by literature.

Response: Comments are kindly noted.

[f.] Avoid recommendations in the paper.

Response: Recommendations have been removed/reviewed

[g.] Specific comments - Line 12-15: delete sentence

Response: Sentence has been deleted.

[h.] Line 22-24: delete sentence

Response: The recommendation in this section has been deleted and the first sentence edited in compliance with RC1 comments.

[i.] Line 28-33: delete sentence

Response: We saw a need to give an introduction that will enable the reader have a background information about the significance of water to the human body. Sentences have been deleted.

[j.] Line 35: delete sentence Response: This section is already deleted as per RC1 comments.

[k.] Line 58-69: delete sentences

Response: We feel there is a need to state the physical, chemical and biological properties of water. We have summarised the sentences as follows:

"The physical properties of water are colour, turbidity, temperature, taste and odour. The chemical properties of water are pH or the presence of chemicals like arsenic, Iron, Lead, Sodium, Zinc or other toxic organic or inorganic substances. Some chemicals are essential to human and animals in trace amounts, but prolonged exposure in higher amounts can be dangerous to human health. Chemicals can occur naturally from water source; whereas others are as a result of human activities (industrial-mining and human dwellings); agricultural activities (fertilizer and pesticide application); water treatment (supply lines, coagulants); pesticides (public health use); or containing vessels where water is stored (plastic bottles) (WHO, 2017)."

[l.] Line 70-72: delete sentences

Response: Section has been reviewed in accordance with RC1 comments as follows:

"Regulations specifically aim at ensuring the deleterious effects of the chemicals are avoided."

[m.] Line 76-86: delete sentences

Response: Section has been summarised in response to RC1 comments as follows:

"Radiological standards deal with water in contact with radioactivity. Parameters like alpha, beta particles, Radium-226, 228 and Uranium have standards set for drinking water to ensure no untoward health effects to human wellbeing when consumed (Davis, 2010). Radiation occurs naturally (accounting for almost 80% of dose from all radiation sources), from medical diagnosis (accounts for 19.6%) and from man-made sources (accounts for 0.4%) (WHO, 2017)."

[n.] Line 88: Table 1 is not a representative summary of drinking water standards

Response: Comment is kindly noted. Table 1 (see attached Fig. 1) has been updated and where we were able to the bottled water quality has been indicated in parenthesis. In addition, a number of regions adopt drinking water quality standards as bottled water quality standards unless localities institute different regulations.

[o.] Line 101: The = the

Response: correction has been made.

[p.] Line 105: after wards = afterwards

Response: correction has been made.

[q.] Line 121-139: this lacks a critical discussion. Why pH changes over time and why is it of importance? Drinking water standards are not relevant in bottled water:

Response: We had redone this section based on RC1 comments. In addition, we found the analysis applicable since no specific bottled water quality is adopted in the study area/study country. We have further edited it to include comments you have raised:

"The pH values obtained from the tests carried out for the five brands of bottled water are as given in Fig 1. The baseline (day zero) pH values for all brands ranged between 5.25 and 7.4. For the control sample at 14 day and 28 day, there was a general decrease of pH implying increase in acidity in the 5 brands. Brand A showed approximately 3.8% and 5.8% reduction in pH in day 14 and 28; Brand B-3.6% and 7.3%;

Brand C-5.4% and 9.2%; Brand D-13% and 20% (highest reduction); and Brand E-2.8% and 7% respectively. The final pH values at the 28 day for Brands A, B, C, D and E were 6.97, 5.93, 6.45, 4.17 and 6.07 respectively (Fig. 1). Samples exposed to direct sunlight had temperatures of 47°C and 56°C on day 14 and day 28 respectively with Brand A showing 15% and 20% reduction in pH at day 14 and day 28; Brand B-23.4% and 46.7%; Brand C-16% and 39%; Brand D-19.8% and 49%; and Brand E- 27% and 54.8% respectively. The final pH values at the 28 day for Brands A, B, C, D and E were 5.92, 3.41, 4.32, 2.67 and 2.95 respectively. The baseline pH values all fall within range of standard (NSDWQ and US-EPA: 6.5-8.5) except for Brands B and D. For the control samples at the 28 day only Brand A had pH at regulatory level (pH of 6.97). The sources of water may be one of the reasons for the pH values obtained for the control samples. Sample A is sourced directly from a spring aquifer which may explain why it has the lowest pH (Fisher et al., 2017), while samples B-E source water are from deep boreholes which might account for the changes in the values of pH noticed (Wright, 2015). The geology of the locations of water source may also have influence on the pH values. The sample exposed to sunlight had a lower pH (higher acidity) for each Brand and also as the days of exposure to sunlight increased. This is similar to research by (Muhamad et al., 2011), where samples were exposed to sunlight for 5 days (there was a decrease in pH with mean max temperature between 41oC to 47oC). However, the result from this study varies from investigation by (Akharame et al., 2018), where the pH values after the 28 day were all within the WHO and NSDWQ regulation levels. WHO has highlighted health effects of pH<4 to include eye redness and irritation and for pH<2.5 damage to epithelium (WHO, 2003b). The result suggests that sunlight exposure (temperature) affects pH of bottled water."

[r.] Line 148: is = was

Response: correction has been made.

[s.] Line 149: by (Bach et al., 2014) = by Bach et al. (2014)

Response: correction has been made.

[t.] Line 149: not clear. Increase in study of Bach was lower but 140% (which is higher)

Response: That section has been updated to provide a better clarity of expression as follows:

"In that research, water samples had increase in Antimony levels from $0.7\mu$g/L to $0.8\mu$g/L (about 27% or 1.4times the initial value) and $0\mu$g/L to $0.5\mu$g/L after the 10th day of exposing mineral (non-carbonated) and ultrapure water samples to sunlight respectively."

[u.] Line 168-171: delete sentences

Response: sentences have been deleted.

[v.] Line 176: is = was

Response: correction has been made.

[w.] Line 186-201: this lacks a critical discussion. Why nitrate changes over time? Consider deleting

Response: We have kindly retained and updated this section as per response stated in 'item d.'

[x.] Line 205-207: adjust depending on changes in the document

Response: We have kindly retained this section as per responses stated in 'items d. and w.'

[y.] Line 211: adjust depending on changes in the document

Response: We have redone this sentence as thus:

"The level of Nitrate leached as a result of sunlight exposure is within the limit set by NSDWQ and WHO."

[z.] Line 212-218: delete sentences.

Response: Sentences have been deleted.

Thank you.

———————————————————

[Figure]

**Table 1: Regulated Standards of Selected Physical, Chemical, Biological and Radiological Parameters**

| Parameter | NSDWQ[a] | WHO[b] | US EPA[c] [FDA] | EU-DWD[d] | Government of Canada[e] | Suspected Health Effects[a,b,c,d,e] |
|---|---|---|---|---|---|---|
| Turbidity | 5NTU | | NA [5NTU] | 4NTU | 0.3-1NTU | Indicates inefficient water treatment process/distribution system. |
| Temperature | Ambient | - | - | | <15ºC | High temperature enhances microbial growth and problems associated with taste, odor, color and corrosion[a,e] |
| Color | 15TCU | - | [15 color units] | 20mg/L Pt/Co | <15TCU | Indicates unsafe water/presence of chemical or biological impurities. |
| pH | 6.5 – 8.5 | - | 6-8.5 | | 7-10.5 | Acidic pH corrodes conduits & leach chemicals into water, irritate eyes. |
| Antimony (Sb) | - | 20µg/L | 6µg/L [6µg/L] | 5.0µg/L | 6µg/L | Affects blood (increase in cholesterol, decrease in sugar) |
| Arsenic (As) | 0.2mg/L | 10µg/L | [10µg/L] | 10µg/L | 10µg/L [10µg/L] | Affects Skin & nervous system, increases cancer risk |
| Lead (Pb) | 0.01mg/L | 10µg/L | 15µg/L [5µg/L] | 25µg/L | 5µg/L [10µg/L] | Affects central and peripheral nervous system, cancer & kidney damage |
| E.coli | 0cfu/100mL | | 0/100mL [0/100mL] | 0/100mL | 0/100mL [0/100mL] | Gastroenteric disease |
| Total coliforms | 10cfu/100mL | | 0/100mL [0/100mL or <5% incidence/month] | 0/100mL | 0/100mL [10/100mL] | Gastroenteric disease |
| Alpha particles (α)(activity) | 0.1Bq/L | 0.5Bq/L | 15pCi/L [15pCi/L] | | | Increases Cancer risks |
| Beta particles (β) (activity) | 0.1Bq/L | 1Bq/L | 4 millirems per year [4 millirems per year] | | | Increases Cancer risks |
| Radium (Ra)-226 + Radium (Ra)-228 | - | 1Bq/L | 5pCi/L [5pCi/L] | | 0.5Bq/L | Increases Cancer risks |
| Uranium | - | 30µg/L | 30µg/L [30µg/L] | | | Increases Cancer risks |

**Fig. 1.**

---

## Author Response (AR1)

Prof. Luuk Rietveld,
Topical Editor,
DWES Journal.
03/01/2020

Dear Sir,

Letter of Rebuttal: Evaluation of Changes in Some Physico-Chemical Properties of Bottled Water Exposed to Sunlight in Bauchi, Nigeria

We want to appreciate you and our referees for contributing important corrections to the manuscript we submitted for publication. The points you highlighted have enabled us to go through step by step to ensure that we resolve them. Kindly find below our responses/rebuttals to the Referee comments as follows:

1. For Referee One:
    a. Abstract Line 22-24 – I would suggest to leave out the recommendation since more information is needed (especially regarding characterization of the bottles, environmental conditions such as pressure, humidity: : :).
    Response: The recommendation has been removed from the Abstract.
    b. Introduction 1) General observation – Introduction is too long with irrelevant information in this case such as line 70-72 "Some chemicals called disinfection products (DBP) get in water as a result of disinfection (chlorination) in water treatment process. They include the trihalomethanes (THMs) and haloacetic acids (HAAs) which are the main DBPs worth noting (WHO, 2017)."
    Author's Response: The intention of including the paragraph was to inform the readers about the different aspects of water treatment and possible effects of disinfectants on chemical composition of the water. However, the section has been removed and the introduction reviewed.
    c. Sentences should be shorter and to the point. For example: "PET can be recycled by breaking it down into its constituents and using same to make new PET materials, unfortunately large amounts of this product still find its way to landfills, open dumps and improperly disposed waste where it breaks down to micro-plastics and nano-plastics ultimately finding its way to the marine ecosystem with deleterious environmental effects (PETRA, 2015; US EPA, 1995)."
    Response: Sentences have been modified (lines 57-59).
    d. Remove line 35 "Bottled water is a good option compared to other beverages especially those that contain high sugar content."
    Response: Sentence has been removed.
    e. What is the relevance of this paragraph on the topic: "The biological characteristics of water includes presence of pathogenic organisms-viruses, protozoa, helminths, bacteria which can cause illnesses such as typhoid, diarrhoea, tape worms, round worms. The presence of Escherichia coli (E. coli), Enterobacter clocae, Citrobacter freundii, common in stool and sputum of warm-blooded animals including human proves the contamination of water by stool (Salvato et al., 2004; SON, 2007; Weiner and Matthews, 2003; WHO, 2017). Some of the organism however can grow in water distribution systems, reproduce as a result of warm temperatures and be inhaled as aerosols (amoebae Naegleria fowleri and Acanthamoeba spp.) (WHO, 2017). The target for most regulations is to ensure that no pathogenic organism is present in water (FDA, 2016; WHO, 2017).")
    Response: Though the paper focuses on physico-chemical changes in water and no biological test was carried out directly in the study, we believe that the paragraph will inform the readers about the biological aspects of drinking water quality worthy of note. But the paragraph has been removed for lack of relevance to the topic.

f. In the introduction it is mentioned: "Over the course of the year temperatures can rise to as high as 40oC around March-April in the study area. The study was carried out between December, 2019 to January, 2020 when average maximum temperature was 31oC. (World Weather Online, 2020)". However, in the 2.2 paragraph it is mentioned: "December 2019 (Ambient: Max temp = 31oC, Avg. temp = 16oC and Min Temp = 16oC) and January 2020 (Ambient: Max temp = 31oC, Avg. temp = 24oC and Min temp = 14oC) (World Weather Online, 2020) using a destructive sampling technique." Could you explain the difference in average temperatures? Was the temperature continuously monitored during 28 days? How could you explain these data being representative for temperatures of 40C while having experiments done during average of 16 and 24C?
Response: There was error in typing in the average temperature for December 2020 and it has been corrected accordingly (line 113). Secondly, for the water samples the temperature was not continuously measured for 28days, it was measured at the point of sample analysis. Though the study is not representative of temperatures of 40oC, but we believe it is still relevant in understanding the effect of exposing bottled water products to direct sunlight, which is all through the year.

g. Results and discussion: Different bottles either control samples or samples exposed to sunlight are showing changes to pH, antimony, BPA and nitrate concentrations. However, the explanation and link are missing to why is there a different decrease in for example pH. What is the structure of the bottles itself and what is the water matrix? What is affecting one sample to have pH decrease of 3.6 % while other 20%.
Response: There was no study carried out on the structure of the bottles or water matrix to understand and discuss their effect on the pH levels. However, information we have shows that Sample A raw water is sourced from a spring while the other samples (B-E) are sourced from deep boreholes. Depending on the source, dissolution of minerals (for example limestone which tend to increase pH by dissolving carbonates, bicarbonate and hydroxide compounds; or dissolved carbon dioxide which will decrease the pH over time. The process of water treatment and the minerals added can also account for the change of pH in control samples. The discussion of pH results has been modified in the paper (lines 150-161)

2. For Referee Two:
   a. General comments: - Now much emphasis is given to the compliance with EPA and WHO regulations. More important is the effect of PET bottles and sunlight on the quality.
   Response: comments are kindly noted and discussion of results have been adjusted to reflect effect of sunlight on the quality.

   b. The abstract is too long with too specific details on compliance with regulations.
   Response: Abstract has been updated as suggested in addition to RC1 comments.

   c. Introduction has too many generalities, be more specific.
   Response: Introduction has been updated in addition to RC1 comments.

   d. Not clear why pH and nitrate are chosen as indicators for PET leaching. Consider at least removing nitrate as indicator.
   Response: Though pH and Nitrate are not direct indicators for PET leaching, we decided to include them as they are parameters that were indicated on the products and we wanted to study/ensure that they were within acceptable range when exposed to sunlight. In addition, Muhamad et. al. (2011) also postulated that temperatures above 35oC can cause degradation of PET polymers and alter chemical properties (Total

dissolved solids-TDS, $NO_3^-$, $SO_4^{2-}$ and $NH_4^+$ in bottled water. We have kindly included it in discussion of Nitrate results (Included in lines 171-175) as follows:

"The increase in Nitrate can be attributed to the thermal degradation of the polymers in the PET bottle as a result of temperature increase. This is evidenced by research on temperature effects on bottled water by Muhamad et al. (2011), they found that from 35oC 'molecular degradation as a result of overheating' causes chemical parameters (including Nitrate) to be altered. Fig 4 shows temperature of 35oC was exceeded on day 14 and day 28 for samples exposed to sunlight."

We have also updated the sentences line in the manuscript to reflect the chemicals (lines 92-95):

"This study seeks to investigate the changes in levels of specific parameters (Antimony, Bisphenol A, Nitrates, pH) in five (5) commercially available market brands of bottled water continually exposed to sunlight for 0 – 28days in Bauchi, Bauchi State and the effects on the quality of the water."

e. Results frequently lacks critical discussion on mechanisms supported by literature.
Response: Comments are kindly noted. We have included discussions to support mechanisms for pH, Nitrate and adjusted discussions in relevant sections (lines 109-113 for pH, 131-139 for Antimony, 171-175 for Nitrate)

f. Avoid recommendations in the paper.
Response: Recommendations have been removed/reviewed in the abstract (line 20-22)

g. Specific comments - Line 12-15: delete sentence
Response: Sentence has been deleted.

h. Line 22-24: delete sentence
Response: The recommendation in this section has been deleted and the first sentence edited in compliance with RC1 comments.

i. Line 28-33: delete sentence
Response: We saw a need to give an introduction that will enable the reader have a background information about the significance of water to the human body. However, sentences have been deleted.

j. Line 35: delete sentence
Response: This section is already deleted as per RC1 comments.

k. Line 58-69: delete sentences
Response: The biological properties were removed in accordance with RC1 comments. We feel there is a need to state the physical, and chemical properties of water. We have summarised the sentences as follows (lines 61-69):

"The physical properties of water are colour, turbidity, temperature, taste and odour. The chemical properties of water are pH or the presence of chemicals like arsenic, Iron, Lead, Sodium, Zinc or other toxic organic or inorganic substances. Some chemicals are essential to human and animals in trace amounts, but prolonged exposure in higher

amounts can be dangerous to human health. Chemicals can occur naturally from water source; whereas others are as a result of human activities (industrial-mining and human dwellings); agricultural activities (fertilizer and pesticide application); water treatment (supply lines, coagulants); pesticides (public health use); or containing vessels where water is stored (plastic bottles) (WHO, 2017)."

l. Line 70-72: delete sentences
Response: Section has been reviewed in accordance with RC1 comments as follows:

"Regulations specifically aim at ensuring the deleterious effects of the chemicals are avoided."

m. Line 76-86: delete sentences
Response: The biological characteristics were deleted as per RC1 comments.

n. Line 88: Table 1 is not a representative summary of drinking water standards
Response: Comment is kindly noted. Table 1 (see attached Fig. 1) has been updated and where we were able to the bottled water quality has been indicated in parenthesis. In addition, a number of regions adopt drinking water quality standards as bottled water quality standards unless localities institute different regulations. For example, in the study area there are no standards for bottled water quality which is similar to details in an article by Dr. Alan A. Leff (2000) (https://www.wqpmag.com/bottled-water-quality-guidelines-fda-who-or-what). The paragraph in line 88 has been edited:
"A summary of the drinking water and bottled water (shown in parenthesis) quality standards available as issued by major institutions for physical, chemical, biological and radiological characteristics; and suspected health effects are presented in Table 1. As seen in Table 1, in US and Canada as with many other places a number of the regulations cover for both drinking water and bottled water quality while for regions like Nigeria no bottled water regulation is in place and sometimes drinking water quality standards are adopted."

Table 1: Regulated Standards of Selected Physical, Chemical, Biological and Radiological Parameters

| Parameter | NSDWQ[a] | WHO[b] | US EPA[c] [FDA] | EU-DWD[d] | Government of Canada[e] | Suspected Health Effects[a,b,c,d,e] |
|---|---|---|---|---|---|---|
| Turbidity | 5NTU | | NA [5NTU] | 4NTU | 0.3-1NTU | Indicates inefficient water treatment process/distribution system. |
| Temperature | Ambient | - | - | | <15°C | High temperature enhances microbial growth and problems associated with taste, odor, color and corrosion[b,c] |
| Color | 15TCU | - | [15 color units] | 20mg/L Pt/Co | <15TCU | Indicates unsafe water/presence of chemical or biological impurities. |
| pH | 6.5 – 8.5 | - | 6-8.5 | | 7-10.5 | Acidic pH corrodes conduits & leach chemicals into water, irritate eyes. |
| Antimony (Sb) | - | 20µg/L | 6µg/L [6µg/L] | 5.0µg/L | 6µg/L | Affects blood (increase in cholesterol, decrease in sugar) |
| Arsenic (As) | 0.2mg/L | 10µg/L | [10µg/L] | 10µg/L | 10µg/L [10µg/L] | Affects Skin & nervous system, increases cancer risk |
| Lead (Pb) | 0.01mg/L | 10µg/L | 15µg/L [5µg/L] | 25µg/L | 5µg/L [10µg/L] | Affects central and peripheral nervous system, cancer & kidney damage |
| E. coli | 0cfu/100mL | | 0/100mL [0/100mL] | 0/100mL | 0/100mL [0/100mL] | Gastroenteric disease |
| Total coliforms | 10cfu/100mL | | 0/100mL [0/100mL or <5% incidence/month] | 0/100mL | 0/100mL [10/100mL] | Gastroenteric disease |
| Alpha particles (α)(activity) | 0.1Bq/L | 0.5Bq/L | 15pCi/L [15pCi/L] | | | Increases Cancer risks |
| Beta particles (β) (activity) | 0.1Bq/L | 1Bq/L | 4 millirems per year [4 millirems per year] | | | Increases Cancer risks |
| Radium (Ra)-226 + Radium (Ra)-228 | - | 1Bq/L | 5pCi/L [5pCi/L] | | 0.5Bq/L | Increases Cancer risks |
| Uranium | - | 30µg/L | 30µg/L [30µg/L] | | | Increases Cancer risks |

o. Line 101: The = the
   Response: correction has been made.

p. Line 105: after wards = afterwards
   Response: correction has been made.

q. Line 121-139: this lacks a critical discussion. Why pH changes over time and why is it of importance? Drinking water standards are not relevant in bottled water:
   Response: We had redone this section based on RC1 comments. In addition, we found the analysis applicable since no specific bottled water quality is adopted in the study area/study country. We have further edited it to include comments you have raised (lines 141-161):

   "The pH values obtained from the tests carried out for the five brands of bottled water are as given in Fig 1. The baseline (day zero) pH values for all brands ranged between 5.25 and 7.4. For the control sample at 14 day and 28 day, there was a general decrease of pH implying increase in acidity in the 5 brands. Brand A showed approximately 3.8% and 5.8% reduction in pH in day 14 and 28; Brand B-3.6% and 7.3%; Brand C-5.4% and 9.2%; Brand D-13% and 20% (highest reduction); and Brand E- 2.8% and 7% respectively. The final pH values at the 28 day for Brands A, B, C, D and E were 6.97, 5.93, 6.45, 4.17 and 6.07 respectively (Fig. 1). Samples exposed to direct sunlight had temperatures of 47°C and 56°C on day 14 and day 28 respectively with Brand A showing 15% and 20% reduction in pH at day 14 and day 28; Brand B-23.4% and 46.7%; Brand C-16% and 39%; Brand D-19.8% and 49%; and Brand E- 27% and 54.8% respectively. The final pH values at the 28 day for Brands A, B, C, D and E were 5.92, 3.41, 4.32, 2.67 and 2.95 respectively.
   The baseline pH values all fall within range of standard (NSDWQ and US-EPA: 6.5-8.5) except for Brands B and D. For the control samples at the 28 day only Brand A had pH at regulatory level (pH of 6.97). The sources of water may be one of the reasons for the pH values obtained for the control samples. Sample A is sourced directly from a spring aquifer which may explain why it has the lowest pH (Fisher et al., 2017), while samples B-E source water are from deep boreholes which might account for the changes in the values of pH noticed (Wright, 2015). The geology of the locations of water source may also have influence on the pH values.
   The sample exposed to sunlight had a lower pH (higher acidity) for each Brand and also as the days of exposure to sunlight increased. This is similar to research by (Muhamad et al., 2011), where samples were exposed to sunlight for 5 days (there was a decrease in pH with mean max temperature between 41oC to 47oC). However, the result from this study varies from investigation by (Akharame et al., 2018), where the pH values after the 28 day were all within the WHO and NSDWQ regulation levels. WHO has highlighted health effects of pH<4 to include eye redness and irritation and for pH<2.5 damage to epithelium (WHO, 2003b). The result suggests that sunlight exposure (temperature) affects pH of bottled water."

r. Line 148: is = was
   Response: correction has been made.

s. Line 149: by (Bach et al., 2014) = by Bach et al. (2014)
   Response: correction has been made.

t.  Line 149: not clear. Increase in study of Bach was lower but 140% (which is higher)
    Response: We realise that the statement did not clearly reflect what we intended. We apologise and have kindly updated section has been updated to provide a better clarity of expression as follows (lines 1710173):

    "In that research, water samples had increase in Antimony levels from 0.7µg/L to 0.98µg/L (about 27% or 1.4times the initial value) and 0µg/L to 0.5µg/L after the 10th day of exposing mineral (non-carbonated) and ultrapure water samples to sunlight respectively."

u.  Line 168-171: delete sentences
    Response: sentences have been deleted.

v.  Line 176: is = was
    Response: correction has been made.

w.  Line 186-201: this lacks a critical discussion. Why nitrate changes over time? Consider deleting
    Response: We have kindly retained and updated this section as per response stated in 'item d.'

x.  Line 205-207: adjust depending on changes in the document
    Response: We have kindly retained this section as per responses stated in 'items d. and w.'

y.  Line 211: adjust depending on changes in the document
    Response: We have redone this sentence as thus (line 246):

    "The level of Nitrate leached as a result of sunlight exposure is within the limit set by NSDWQ and WHO."

z.  Line 212-218: delete sentences.
    Response: Sentences have been deleted.

We sincerely thank you and our referees for the time invested in making the manuscript better and for graciously considering it for publishing. We are available for any further corrections as you may require to make the manuscript publishable.

Thank you.

Dr Rose E. Daffi and Fwangmun B. Wamyil